# Epidemiology of Human Parainfluenza Virus Infections among Pediatric Patients in Hainan Island, China, 2021–2023

**DOI:** 10.3390/pathogens13090740

**Published:** 2024-08-30

**Authors:** Meifang Xiao, Afreen Banu, Xiangyue Zeng, Shengjie Shi, Ruoyan Peng, Siqi Chen, Nan Ge, Cheng Tang, Yi Huang, Gaoyu Wang, Xiaoyuan Hu, Xiuji Cui, Jasper Fuk-Woo Chan, Feifei Yin, Meng Chang

**Affiliations:** 1Department of Clinical Laboratory, Center for Laboratory Medicine, Hainan Women and Children’s Medical Center, Hainan Medical University, Haikou 570206, China; meifang@lincoln.edu.my (M.X.); zxy865673000@163.com (X.Z.); 15874843023@163.com (S.S.); 2Department of Microbiology, Faculty of Medicine, Lincoln University College, Petaling Jaya 47301, Malaysia; afreen@lincoln.edu.my; 3Hainan Medical University-The University of Hong Kong Joint Laboratory of Tropical Infectious Diseases, Key Laboratory of Tropical Translational Medicine of Ministry of Education, School of Basic Medicine and Life Sciences, Hainan Medical University, Haikou 571199, China; pry0302@163.com (R.P.); qisi49u@163.com (S.C.); gn18689738690@163.com (N.G.); tsc1104@163.com (C.T.); huangyi930925@163.com (Y.H.); wgy1205@hainmc.edu.cn (G.W.); 15300213416@163.com (X.H.); cuixj26@163.com (X.C.); 4Department of Pathogen Biology, Hainan Medical University, Haikou 571199, China; 5State Key Laboratory of Emerging Infectious Diseases, Department of Microbiology, and Carol Yu Centre for Infection, Li Ka Shing Faculty of Medicine, The University of Hong Kong, Pokfulam, Hong Kong SAR 999077, China; jfwchan@hku.hk; 6Department of Infectious Diseases and Microbiology, The University of Hong Kong-Shenzhen Hospital, Shenzhen 518053, China

**Keywords:** human parainfluenza virus, COVID-19 pandemic, pediatric patients, epidemiology, Hainan Island

## Abstract

Human parainfluenza viruses (HPIVs) are the leading causes of acute respiratory tract infections (ARTIs), particularly in children. During the COVID-19 pandemic, non-pharmaceutical interventions (NPIs) significantly influenced the epidemiology of respiratory viruses. This study analyzed 19,339 respiratory specimens from pediatric patients with ARTIs to detect HPIVs using PCR or tNGS, focusing on the period from 2021 to 2023. HPIVs were identified in 1395 patients (7.21%, 1395/19,339), with annual detection rates of 6.86% (303/4419) in 2021, 6.38% (331/5188) in 2022, and 7.82% (761/9732) in 2023. Notably, both the total number of tests and HPIV-positive cases increased in 2023 compared to 2021 and 2022. Seasonal analysis revealed a shift in HPIV prevalence from winter and spring in 2021–2022 to spring and summer in 2023. Most HPIV-positive cases were in children aged 0–7 years, with fewer infections among those aged 7–18 years. Since June 2022, HPIV-3 has been the most prevalent serotype (59.55%, 524/880), whereas HPIV-2 had the lowest proportion (0.80%, 7/880). The proportions of HPIV-1 (24.89%, 219/880) and HPIV-4 (15.45%, 136/880) were similar. Additionally, the incidence of co-infections with other common respiratory pathogens has increased since 2021. This study highlights rising HPIV detection rates post-COVID-19 and underscores the need for continuous surveillance of HPIVs to inform public health strategies for future epidemic seasons.

## 1. Introduction

During the COVID-19 pandemic, a virological anomaly that garnered attention was the persistent prevalence of human parainfluenza viruses (HPIVs) [1,2]. The non-pharmaceutical interventions (NPIs) implemented to mitigate SARS-CoV-2 transmission—such as mask-wearing, social distancing, and lockdowns—resulted in a substantial reduction in the incidence of respiratory viruses, including influenza virus, respiratory syncytial virus (RSV), and human metapneumovirus (hMPV) [3,4,5]. However, in notable contrast, HPIV demonstrated adaptability and appeared to coexist with SARS-CoV-2 at a rate consistent with its typical seasonal patterns despite the decrease in interpersonal contacts due to these public health measures [6]. HPIV is well-known for its efficient transmission through respiratory droplets, which likely played a pivotal role in the sustained propagation of HPIV within populations, especially when pandemic measures such as masking, social distancing, and lockdowns were not in place [7]. The ability of HPIV to maintain its prevalence during the COVID-19 pandemic may be attributed to several factors, including its transmission through respiratory droplets, which might have continued despite the widespread implementation of NPIs. These measures may not have been fully effective in preventing the spread of HPIV, particularly in environments where young children were in close contact, such as in daycares and schools. Additionally, the virus’s resilience in various environmental conditions, especially in tropical climates like Hainan Island, could have supported its ongoing transmission. High humidity and warm temperatures are known to influence the survival and spread of respiratory viruses. Moreover, the potential for HPIV to evade immune responses or take advantage of reduced immunity due to decreased exposure to other respiratory viruses during the pandemic may have further facilitated its continued circulation [8,9,10].

HPIVs are enveloped, non-segmented, negative-sense single-stranded RNA viruses belonging to the *Paramyxoviridae* family [7]. There are four major serotypes of HPIV: HPIV-1, HPIV-2, HPIV-3, and HPIV-4 [7]. HPIV is one of the major viral causes of upper respiratory tract infection (URI) and lower respiratory tract infection (LRI) in children, accounting for about 10% of hospitalized children with respiratory tract infections. HPIV-1 and HPIV-2 are best known as the primary causes of croup, whereas HPIV-3 is a common cause of bronchiolitis and pneumonia [11,12]. HPIV-4 infection has a low prevalence [7]. HPIV is a significant cause of morbidity and mortality worldwide, particularly among children in developing countries [12,13]. It is also an important cause of nosocomial infection, which can be life-threatening in certain individuals, such as transplant or immunocompromised patients [14,15]. Moreover, HPIVs often co-circulate with other respiratory viruses, such as RSV and Influenza virus, complicating diagnosis and clinical management [16]. Co-infections can exacerbate the severity of respiratory symptoms and may lead to worse clinical outcomes [17]. Despite concerted efforts, epidemiologic studies of HPIV have proven challenging [18]. There is currently no available vaccine or specific antiviral treatment for HPIV infection [19,20]. Consequently, further research is imperative to understand the epidemiological features of HPIVs.

In response to the COVID-19 outbreak in early 2020, China implemented strict public health measures, culminating in the “Dynamic Zero-COVID” policy by August 2021 to combat the Delta variant [21]. This policy, which involved extensive lockdowns, mask mandates, and widespread testing, remained until December 2022, significantly reducing the transmission of respiratory viruses by limiting interpersonal contact [22]. In March and August 2022, Hainan experienced two major COVID-19 waves, prompting rigorous enforcement of non-pharmaceutical interventions (NPIs) during these months, which we refer to as the “strict NPI period.” These NPIs were rigorously maintained until the policy’s repeal in December 2022. The implementation of these NPIs also affected the transmission of other respiratory viruses, including respiratory syncytial virus, and potentially influenced pediatric HPIV epidemics. Analyzing HPIV infection trends before and after the pandemic could offer valuable insights for future prevention and control in children [23]. 

This study investigates the epidemiological shifts in HPIV infections among children on the Hainan Islands, focusing on the periods before and after the termination of the zero-COVID policy. Hainan, a tropical island located at the southernmost tip of China, is known for its warm temperatures, high humidity, and status as a major free trade port and tourist destination [24]. These unique climatic and geographical attributes potentially present distinct health challenges. Specifically, the research provides insights into how policy changes, notably the ending of the zero-COVID policy, influenced the dynamics of respiratory viruses from 2021 to 2023. The aim of this investigation is to enhance our comprehension of HPIV circulation and associated diseases.

## 2. Materials and Methods

### 2.1. Patient Enrollment and Data Collection

This study was based on the surveillance of respiratory pathogens in hospitalized children. It included 19,339 pediatric patients (aged ≤18 years) diagnosed with acute lower respiratory tract infections (ARTIs) and who were admitted to the Hainan Maternal and Child Health Hospital from January 2021 to December 2023. Hainan Maternal and Child Health Hospital, located in Hainan Province, China, is a specialized institution dedicated to maternal and child healthcare. As one of the largest pediatric care centers on Hainan Island, the hospital is equipped with advanced diagnostic facilities, including PCR and next-generation sequencing (NGS) technologies, which were utilized in this study. The hospital serves a diverse population, providing a broad representation of pediatric respiratory infections on the island, which makes it an ideal setting for this epidemiological study. The inclusion criteria for the participants were as follows: (1) aged 18 or younger, (2) hospitalized during the study period, (3) diagnosed with ARTIs (symptoms such as fever, headache, rhinorrhea, tachypnea, cough, sore throat, sputum, chest pain, or radiologic evidence indicative of pneumonia or bronchitis), and (4) with complete clinical data. Necessary data for eligible participants included demographic information, clinical features, clinical outcomes, treatment courses, and laboratory test results, which were retrieved from the hospital information system and the laboratory information management system.

### 2.2. Collection of Samples and Nucleic Acid Extraction

Nasopharyngeal swabs were collected upon admission following established clinical procedures, with consent obtained from all participants. The swabs were then suspended in a specified sample preservation medium. Specimens intended for short-term testing were refrigerated at 2–8 °C, while those designated for further testing were preserved in a −80 °C freezer. Nucleic acid extraction was performed using a nucleic acid purification kit (Cat# KS118-BYTQ-96) from KingCreate Biotechnology (Guangzhou, China).

### 2.3. Detection of Pathogens

During this study period (January 2021 to December 2023), the respiratory pathogen assay was progressively improved. From January 2021 to May 2022, the assay was used for the multiplex PCR combined with capillary electrophoresis analysis for only 18 common respiratory pathogens provided by KingMed Diagnostics (Guangzhou, China), including RSV, rhinovirus, adenovirus, human parainfluenza virus, coronavirus, influenza virus A, influenza virus A pdmH1N1(2009), bocavirus, *Mycoplasma pneumoniae*, influenza B virus, parvovirus, influenza virus A H3N2, *Chlamydia*, *Haemophilus influenzae*, *Streptococcus pneumoniae*, *Legionella pneumophila*, *Mycobacterium tuberculosis*, and *Bordetella pertussis* [25,26]. From June 2022 to December 2023, the assay was converted to targeted next-generation sequencing (tNGS) provided by KingMed Diagnostics (Guangzhou, China). The targeted next-generation sequencing (tNGS) panel employed in our study was meticulously designed to focus on the conserved genomic regions of 153 pathogens, covering over 95% of known respiratory infections. This comprehensive panel demonstrated its capability by detecting a wide range of pathogens, including 65 bacterial species, 68 viruses (of which 25 were DNA viruses and 43 were RNA viruses), 14 fungi, and 6 mycoplasmas/chlamydiae [27,28]. The generated sequencing read data underwent a bioinformatics process that included multiple steps as follows: quality control, database alignment, microorganism identification, and the generation of a reliable pathogen report. To identify positive signals for specific pathogens, the sequence reads matching the targeted species were counted and normalized to reads per 100,000 reads (RPhK). Samples with specific RPhK values of ≥10 were considered positive for the presence of the corresponding pathogen, while those with lower values were reported as ‘absent’.

### 2.4. Statistical Analysis

The data were analyzed using the IBM SPSS Statistics program (version 26). The data were analyzed using the χ^2^ test. *p*-values less than 0.05 were considered statistically significant. GraphPad Prism 6.0 software was used for the statistical analysis of graphs.

## 3. Results

### 3.1. Characteristics and HPIV Detection of Overall Samples

From January 2021 to December 2023, a total of 19,339 pediatric inpatients diagnosed with ARTIs were admitted to Hainan Maternal and Child Health Hospital. The demographic details of the overall ARTI cases have been summarized in Table 1. A total of 19,339 pediatric patients hospitalized for acute respiratory tract infections (ARTIs) were included in the study (4419 in 2021, 5188 in 2022, and 9732 in 2023), all of whom were tested for HPIV. Of these, 1395 tested positive for HPIV. Overall, gender significantly affected the data across all years (*p* < 0.05), with the proportion of males generally higher than that of females. Specifically, in 2021, gender did not significantly affect the data (*p* > 0.05), with similar proportions of males and females. In 2022, gender significantly impacted the data (*p* < 0.05), with the proportion of males notably higher than that of females. In 2023, gender also significantly impacted the data (*p* < 0.05), with the proportion of males notably higher than that of females. Age significantly impacted the data across all years (*p* < 0.05), with notable differences in the proportions among different age groups. In 2021, age significantly impacted the data (*p* < 0.05), with notable differences in the proportions among different age groups. In 2022, age significantly impacted the data (*p* < 0.05), with notable differences in the proportions among different age groups. In 2023, age significantly impacted the data (*p* < 0.05), with notable differences in the proportions among different age groups.

The season had a significant impact on the overall data (*p* < 0.05) across all years, with notable differences in the proportions among different seasons. In 2021, season significantly impacted the data (*p* < 0.05), with notable differences in the proportions among different seasons, particularly with spring and winter having higher proportions. In 2022, the season significantly impacted the data (*p* < 0.05), with notable differences in the proportions among different seasons, particularly with winter having the highest proportion. In 2023, the season significantly impacted the data (*p* < 0.05), with summer having the highest proportion.

In summary, gender, age, and season all had significant effects on the overall data across all years (*p* < 0.05), with the proportion of males being generally higher than that of females, as well as notable differences in the proportions among different age groups, and variations in the proportions among different seasons (Table 1).

### 3.2. Prevalence of HPIV among Pediatric Patients in Hainan from 2021 to 2023

To assess the impact of the COVID-19 pandemic on children infected with HPIV, we analyzed the epidemiological data from the Hainan Women and Children’s Medical Center spanning from January 2021 to December 2023. The overall annual trends and the monthly distribution of positive cases are depicted in Figure 1A. Compared to the years significantly affected by COVID-19 (2021 and 2022), both the total number of tests conducted and the number of HPIV-positive cases increased in 2023. The positivity rate for HPIV in 2023 was 7.82% (761/9732), markedly higher than in 2021 with 6.86% (303/4419) and 2022 with 6.38% (331/5188). During 2021–2022, the peak periods of HPIV infections were January, April to June, and December of 2021, and January to April and November to December of 2022. By 2023, the peak infection periods had returned to the pre-pandemic norm of April to July, as described in the literature. 

Since June 2022, the tNGS method has been employed to detect HPIV. This methodological change enabled the specific identification of serotypes (HPIV-1–4). Consequently, we conducted a further subtype analysis of the HPIV data from June 2022 to December 2023 (Figure 1A and Figure 2B). A total of 880 positive cases were identified during this period, with the numbers of positive cases for HPIV-1, HPIV-2, HPIV-3, and HPIV-4 being 219 (24.89%), 7 (0.80%), 524 (59.55%), and 136 (15.45%), respectively. The data reveal that from June to July 2022 (summer), HPIV-1 and HPIV-3 were the predominant serotypes causing infections. From October to December 2022 (late autumn to early winter), HPIV-4 emerged as the predominant infecting serotype. Subsequently, from March to September 2023 (spanning spring, summer, and early autumn), HPIV-3 regained its predominance. In the period from October to December 2023 (late autumn to early winter), HPIV-1, HPIV-3, and HPIV-4 were all major serotypes responsible for infections. Overall, HPIV-3 exhibited a year-round presence, with a notable peak in infections during early summer and early autumn (April to July). HPIV-4 mainly caused concentrated infections in late autumn to early winter (October to December). HPIV-1 showed peaks in infections during the summer (April to June) and late autumn to early winter (October to December). Due to the relatively low number of positive cases, HPIV-2 did not exhibit a clear seasonal epidemic pattern.

In 2022, Hainan encountered two significant COVID-19 waves in April and August. Subsequently, NPIs were implemented and maintained across the year until the repeal of the zero-COVID policies. Due to these measures, a notable suppression of HPIV infections was observed, with the infection rates exhibiting a significant decrease in April and August 2022.

Figure 1B outlines the seasonality of HPIV infections. It is evident that HPIV susceptibility was higher during the spring and winter months of 2021–2022, whereas spring and summer became the seasons with higher positivity rates in 2023, consistent with descriptions in the literature. These results suggest that the COVID-19 pandemic has had a significant impact on the epidemiological characteristics of HPIV infections in children.

### 3.3. Epidemiology of HPIV in Pediatric Patients across Different Age Groups

Furthermore, we analyzed the age-related susceptibility to HPIV infection by categorizing participants into four age groups. Infection rates were highest in the 0–1 year and 1–3 year groups, at 8.56% (506/5910) and 9.72% (485/4989), respectively, followed by the preschool group, with the lowest rates observed in the school-age group. The differences between the groups are statistically significant (χ^2^ = 185.885, *p* = 0.000). (Table 1). HPIV-positive cases were predominantly found in children aged 0–7 years, with fewer infections noted during adolescence (7–18 years), indicating that children in the 0–7 years old bracket may be more susceptible to HPIV. Following the COVID-19 pandemic years (2021–2022), an increase in HPIV positivity rates was observed in 2023 across all age groups (except for the 3–7 year group), suggesting that the pandemic may also have influenced the age distribution of HPIV infections (Figure 2A).

Regarding the correlation between different serotypes and age groups, as illustrated in Figure 2B, HPIV-3 was more susceptible in children aged 0–3 years, while HPIV-1 and HPIV-4 showed the highest positivity rates in the 3–7 years group. Due to the extremely low overall positivity rate of HPIV-2, it was challenging to discern any age-related differences within this serotype.

### 3.4. Co-Infections of HPIV and Other Respiratory Pathogens

Lastly, we compiled data on single HPIV infections and mixed infections of HPIV with other common respiratory pathogens from 2021 to 2023, as shown in Figure 3. During this study period, 6972 samples were analyzed using multiplex PCR combined with capillary electrophoresis. Among these, 507 samples were HPIV-positive, with single HPIV infections accounting for 42.21% (214/507) and mixed infections of HPIV accounting for 57.79% (293/507). In contrast, for samples tested using tNGS, a total of 12,367 samples were analyzed, and 888 samples were HPIV-positive. Among these, single HPIV infections accounted for 6.98% (62/888), and mixed infections of HPIV accounted for 93.02% (826/888).

From the multiplex PCR combined with capillary electrophoresis results, the top five pathogens most frequently co-detected with HPIV were *Streptococcus pneumoniae*, *Haemophilus influenzae*, rhinovirus, adenovirus, and metapneumovirus. Conversely, the tNGS results indicated that the top five pathogens most frequently co-detected with HPIV were rhinovirus, *Streptococcus pneumoniae*, *Haemophilus influenzae*, *Moraxella catarrhalis*, and cytomegalovirus (these data are not provided).

In summary, the co-infection rate detected by tNGS is higher than that detected by the multiplex PCR combined with capillary electrophoresis. Additionally, it was observed that *Streptococcus pneumoniae*, *Haemophilus influenzae*, and rhinovirus are commonly co-infected with HPIV.

## 4. Discussion

This study on the epidemiological shifts in human parainfluenza virus (HPIV) infections among hospitalized children on Hainan Island during the COVID-19 pandemic offers significant insights into how NPIs and policy changes influenced respiratory virus dynamics. The COVID-19 pandemic and the associated NPIs, such as social distancing, mask mandates, and lockdowns, substantially altered the transmission dynamics of respiratory viruses [23,24]. This study observed an increase in HPIV detection rates, rising from 6.86% in 2021 and 6.38% in 2022 to 7.82% in 2023. These detection rates are higher than those reported in Southern China [2], which may be influenced by several factors. First, the regional differences in healthcare infrastructure and diagnostic capabilities could play a role. Hainan Island, being less developed compared to other parts of Southern China, might have experienced delayed or reduced access to healthcare services during the pandemic, leading to a buildup of cases that were later detected as the healthcare system normalized [29]. Moreover, the unique geographic and climatic conditions of Hainan Island might have contributed to a more sustained transmission of HPIV during and after the pandemic. The island’s tropical climate may facilitate the year-round circulation of respiratory viruses, differing from the more seasonal patterns observed in other regions of China [30]. The differences in the HPIV detection rates among pediatric patients are likely influenced by varying economic and healthcare conditions, with the positivity rates being significantly lower in developed regions compared to developing regions [16]. As healthcare conditions improve in regions like Hainan, the disparity in detection rates may decrease. The initial suppression of HPIV transmission during the strict NPI period highlighted the effectiveness of these measures in controlling respiratory infections. However, as the NPIs were relaxed from 2021 to 2023, the subsequent increase in HPIV cases illustrates the virus’s resilience and ability to re-establish transmission.

The HPIV’s primary mode of transmission through respiratory droplets likely allowed it to continue spreading in environments where close contact persisted, such as in households and daycare centers, despite broader social distancing measures. Unlike other respiratory viruses like influenza and RSV, which experienced marked declines, HPIV’s efficient transmission mechanisms may have enabled it to maintain its prevalence. Additionally, HPIV might possess immune evasion capabilities that allow it to avoid detection and clearance by the host immune system, further supporting its persistence during periods of reduced human interaction. Moreover, the concept of “immunity debt”—where reduced exposure to common pathogens during the pandemic led to lower population immunity—may have made individuals, particularly children, more susceptible to infection as the NPIs were lifted [10]. This underscores the importance of understanding HPIV’s unique transmission dynamics and immune evasion strategies.

Additionally, HPIV-1 and HPIV-2 are closely associated with croup, a significant respiratory condition in young children. The association between HPIV and croup emphasizes the importance of early diagnosis and targeted management to prevent severe outcomes in pediatric populations [7]. The resilience of HPIV during the pandemic, despite a significant reduction in other respiratory viruses, highlights the need for continued clinical awareness and preparedness as NPIs are relaxed and respiratory viruses re-emerge. First, HPIV is transmitted primarily through respiratory droplets, a mode of transmission that may have allowed it to persist, even under stringent public health measures like masking and social distancing [7]. Unlike other viruses, such as influenza and RSV, which saw marked declines during the pandemic, HPIV’s efficient transmission mechanisms likely contributed to its continued prevalence. Additionally, HPIV might possess immune evasion capabilities that allow it to avoid detection and clearance by the host immune system, further supporting its persistence during periods of reduced human interaction [16]. This characteristic could explain why HPIV remained relatively stable while other respiratory viruses were suppressed by NPIs. Moreover, the phenomenon of ‘immunity debt’—where reduced exposure to common pathogens during the pandemic led to lower population immunity—may have made individuals, particularly children, more susceptible to HPIV infections as the NPIs were lifted. Therefore, understanding the unique transmission dynamics and immune evasion strategies of HPIV is crucial for developing effective public health responses to future outbreaks.

The seasonal analysis revealed a shift in HPIV prevalence from winter and spring in 2021–2022 to spring and summer in 2023. Similar studies have found that prior to the COVID-19 pandemic, the primary epidemic seasons for HPIV were spring and summer. However, following the pandemic, the detection rate of HPIV infections peaked in December 2020 and January 2021 [31]. This change in seasonal patterns suggests that HPIV transmission adapted to the post-pandemic environment. The alteration in peak infection periods could be due to several factors, including changes in human behavior, climatic conditions, and the population’s immunity landscape [8,32,33]. During the pandemic, reduced exposure to respiratory viruses might have lowered immunity in the population, making children more susceptible when the NPIs were lifted [8,32,34].

The age distribution of HPIV-positive cases showed that the majority were in children aged 0–7 years, with the highest infection rates in the infant (0–1 years) and toddler (1–3 years) groups at 8.56% and 9.72%, respectively. This age-related susceptibility aligns with previous research showing that younger children are more prone to respiratory infections due to their still-developing immune systems and increased exposure in group settings such as daycare centers and schools [33,35].

Previous research on pediatric HPIV infections indicated a concentration of HPIV-positive cases among children aged 3 to 7 years in Henan, located in central China [31]. A notable survey conducted in the southern regions of China, alongside several other reports, has demonstrated a higher susceptibility to HPIV infection among children under the age of three. However, in our study, the increase in HPIV positivity rates across all age groups in 2023, except for the 3–7 years group, suggests that children in this age group may have developed partial immunity due to prior exposures, potentially offering some protection against HPIV. Additionally, this age group might have experienced more typical social interactions earlier during the pandemic recovery, which could have stabilized these particular infection rates compared to younger children who had less exposure during the critical developmental years. This study’s findings similarly indicate that the HPIV infection rate in children below three years is significantly higher compared to other age groups. These observed disparities in HPIV infection rates imply that in addition to immune system development, geographic and climatic factors may also contribute [17,36,37,38]. HPIV-3 was identified as the most prevalent serotype, particularly in children aged 0–3 years. HPIV-3 is known for causing severe lower respiratory tract infections, such as bronchiolitis and pneumonia, which are more common in younger children [38,39,40]. This study’s findings suggest that HPIV-3 may have a higher transmission potential or cause more severe disease in this age group [41]. Conversely, HPIV-2 had the lowest infection rate and did not show a clear seasonal pattern, indicating a lower transmission potential or milder infections that do not require hospitalization as frequently [39].

This study also reports an increase in co-infections with other common respiratory pathogens since 2021. The application of targeted next-generation sequencing (tNGS) in this study was crucial for identifying multiple pathogens involved in co-infections. This comprehensive diagnostic approach can significantly guide the development of more effective treatment strategies. The increased incidence of co-infections underscores the need for comprehensive diagnostic testing in managing respiratory infections, particularly in pediatric and immunocompromised populations. By identifying the full spectrum of pathogens present, healthcare providers can tailor treatment regimens to address all underlying infections, potentially improving clinical outcomes. It is well known that *Streptococcus pneumoniae* and *Moraxella catarrhalis* are common nasopharyngeal colonizers in children, but they can become pathogenic, especially during viral infections like HPIV. Viral-induced disruption of mucosal barriers can facilitate bacterial invasion, leading to more severe respiratory symptoms [42]. In this study, although these bacteria were frequently detected as co-pathogens, their presence alone does not warrant antibiotic treatment unless supported by clinical evidence of bacterial infection. Overuse of antibiotics in cases of mere colonization could contribute to resistance, emphasizing the need for careful clinical assessment before initiating treatment. We believe that the increase in HPIV detection rates, particularly in 2023, can be attributed to a combination of factors. Firstly, the relaxation of COVID-19 preventive measures in 2023, including a reduction in mask-wearing and an overall increase in testing, likely led to higher exposure to respiratory pathogens, resulting in more detected cases. Additionally, as the virulence of SARS-CoV-2 decreased, its restrictive influence on other respiratory pathogens, including HPIV, diminished, allowing for increased circulation of HPIV. While the transition from PCR to targeted next-generation sequencing (tNGS) may have contributed slightly to the higher detection rates, this impact was minimal, as both diagnostic methods were designed to detect all four HPIV serotypes (HPIV-1–4). Therefore, the observed increase in incidence is more likely due to these external factors rather than solely improved diagnostic methods. Additionally, the severity rates of HPIV infections among hospitalized children have been recorded, which were 0.18% (8/4419) in 2021, 0.15% (8/5188) in 2022, and 0.41% (40/9732) in 2023. There are divergent perspectives on the impact of co-infection on clinical severity. Some studies suggest that viral co-infection does not exacerbate the severity [38,43], whereas other research indicates that dual or mixed respiratory virus infections infection may elevate the risk of severe conditions [44,45,46]. Due to the small number of severe cases, it is difficult to establish a link between HPIV infection and the onset of severe illness.

The detection of *Streptococcus pneumoniae* and *Haemophilus influenzae* in co-infections with HPIV warrants a discussion on colonization versus infection. It is important to recognize that these bacteria are common colonizers in the respiratory tract in children, and their presence may not necessarily indicate active infection. Colonization can play a role in the overall respiratory health of the child, potentially influencing susceptibility to viral infections or exacerbating symptoms. Understanding whether these detections represent colonization or active infection is crucial for accurately assessing the clinical implications and guiding appropriate treatment strategies [47].

This study’s findings open several avenues for future research. One area of interest is understanding the mechanisms underlying the resilience and adaptability of HPIV in the face of public health interventions. Further studies could explore the genetic and phenotypic characteristics of HPIV that contribute to its efficient transmission and persistence. Another important research direction is investigating the long-term impact of the COVID-19 pandemic on the epidemiology of other respiratory viruses. Studies could examine how changes in human behavior, immunity, and public health policies influence the transmission dynamics of various respiratory pathogens over time.

This study also possesses several limitations. Firstly, the evaluation of children was confined to a single center with a relatively small patient cohort, limiting the generalizability of our virus-specific clinical correlations to other settings. Additionally, the change in diagnostic methods over time could have influenced the detection rates of various pathogens, potentially affecting the interpretation of trends and patterns observed in our study. Furthermore, this study did not differentiate between the four types of HPIV (HPIV-1–4), nor did it analyze the clinical characteristics associated with bacterial co-infections. Lastly, the lack of long-term follow-up information limits our understanding of the prolonged impact of HPIV infections on pediatric patients.

## 5. Conclusions

In conclusion, the COVID-19 pandemic has profoundly influenced the epidemiology of HPIV infections among hospitalized children on Hainan Island. This study highlights significant increases in the detection rates and changes in seasonal patterns post-pandemic. These findings underscore the importance of continuous surveillance, advanced diagnostic techniques, and adaptive public health strategies to manage the ongoing and future challenges posed by respiratory viruses.

The increased incidence of HPIV, particularly among younger children, and the rise in co-infections with other respiratory pathogens emphasize the need for comprehensive diagnostic approaches and targeted public health interventions. It is also acknowledged that the rise in co-infection rates may be partly due to the enhanced sensitivity of the testing methods, particularly with the adoption of targeted next-generation sequencing (tNGS), alongside the actual epidemiological changes. Future research should focus on understanding the resilience of HPIV, developing effective vaccines and antiviral treatments, and exploring the long-term impacts of the pandemic on respiratory virus epidemiology. Integrating these insights into public health strategies can improve the health outcomes for pediatric populations and other vulnerable groups.

## Figures and Tables

**Figure 1 pathogens-13-00740-f001:**
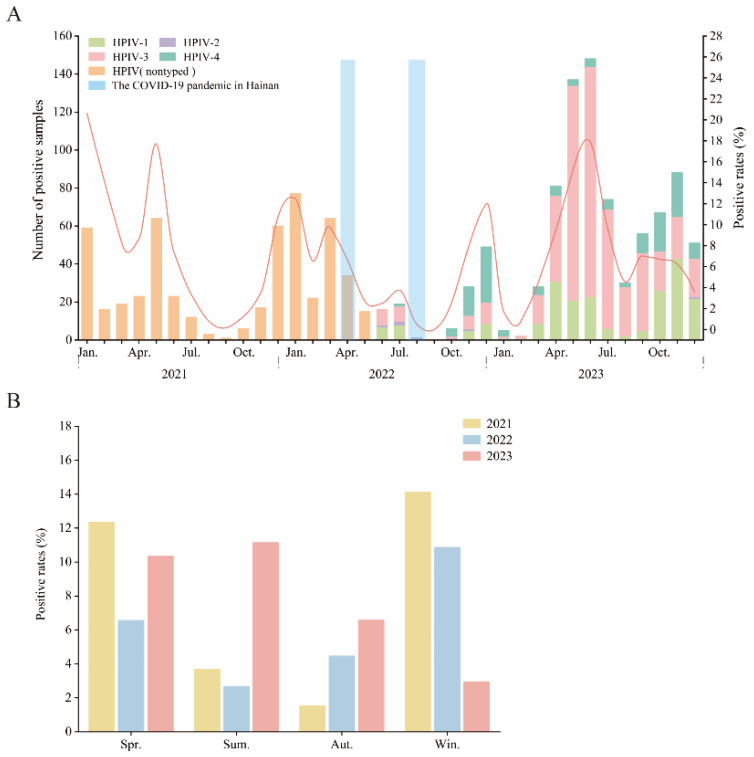
The monthly distribution and seasonality of HPIV infections among hospitalized children in Hainan from 2021 to 2023. (**A**) depicts the overall annual trends and monthly distribution of positive HPIV cases. Since June 2022, the tNGS method has been employed to detect HPIV, enabling the detection and distinction of all four serotypes (HPIV-1, HPIV-2, HPIV-3, and HPIV-4). Each color in the image represents an independent variable, and each column represents a dependent variable. The blue-shaded areas indicate the two epidemic periods of SARS-CoV-2 on Hainan Island. (**B**) shows the seasonality of HPIV infections, with ‘Spr.’, ‘Sum.’, ‘Aut.’, and ‘Win.’ representing the four seasons, respectively. Winter refers to December, January, and February, with March to May in spring, June to August in summer, and September to November in autumn.

**Figure 2 pathogens-13-00740-f002:**
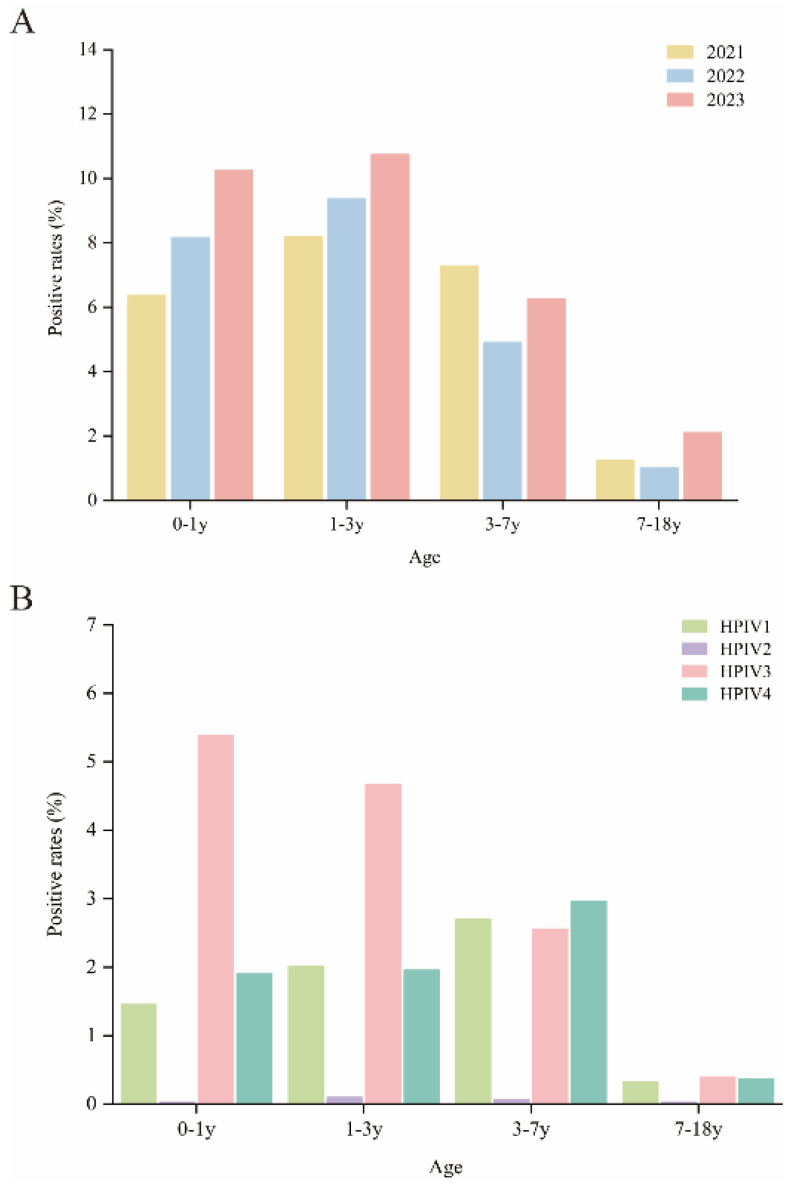
Age distribution of HPIV infections in hospitalized children from 2021 to 2023. Children were categorized into four age groups: the infant group (0–1 years), toddler group (1–3 years), preschool group (3–7 years), and school-age group (7–18 years). (**A**) The age-related susceptibility of HPIV infections. (**B**) The correlation between different serotypes and age groups.

**Figure 3 pathogens-13-00740-f003:**
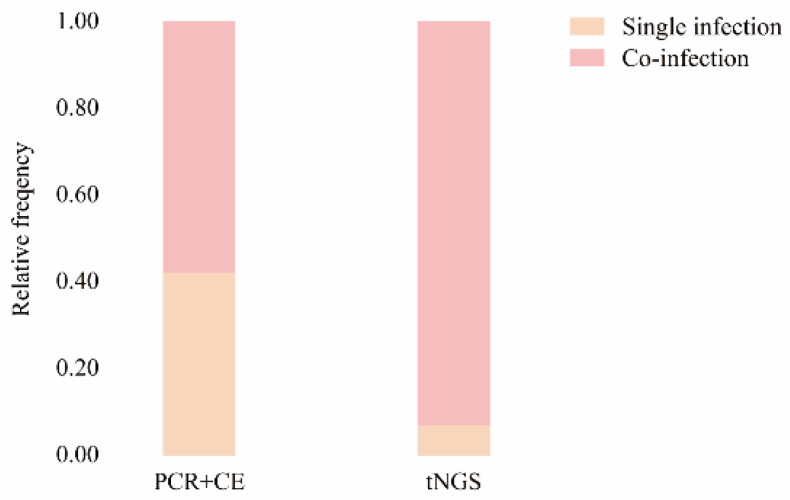
The distribution of HPIV single infection and co-infection among hospitalized children in Hainan from 2021 to 2023. The relative frequency represents the proportion of each part. ‘CE’ stands for ‘capillary electrophoresis’.

**Table 1 pathogens-13-00740-t001:** Characteristics of HPIV infections by gender, age, and season from 2021 to 2023.

Characteristic		Total (*n* = 19,339)	2021 (*n* = 4419)	2022 (*n* = 5188)	2023 (*n* = 9732)
**Gender**	Male	915/11,909 (7.68%)	191/2809 (6.80%)	221/3149 (7.02%)	503/5951 (8.45%)
	Female	480/7430 (6.46%)	112/1610 (6.96%)	110/2039 (5.39%)	258/3781 (6.82%)
χ^2^		10.224	0.039	5.460	8.509
*p*		0.001	0.843	0.019	0.004
Age	0–1	506/5910 (8.56%)	116/1818 (6.38%)	115/1408 (8.17%)	275/2684 (10.25%)
	1–3	485/4989 (9.72%)	107/1306 (8.19%)	121/1292 (9.37%)	257/2391 (10.75%)
	3–7	362/5990 (6.04%)	77/1056 (7.29%)	88/1794 (4.91%)	197/3140 (6.27%)
	7–18	42/2450 (1.71%)	3/239 (1.26%)	7/694 (1.01%)	32/1517 (2.11%)
χ^2^		185.885	16.351	66.865	129.409
*p*		0.000	0.001	0.000	0.000
Season	Spring	463/4937 (9.38%)	106/858(12.35%)	113/1722 (6.56%)	224/2357 (10.35%)
	Summer	325/4629 (7.02%)	38/1033 (3.68%)	36/1344 (2.68%)	251/2252 (11.15%)
	Autumn	266/5491 (4.84%)	24/1571 (1.53%)	34/761 (4.47%)	208/3159 (6.58%)
	Winter	341/4282 (7.96%)	135/957 (14.11%)	148/1361 (10.87%)	58/1964 (2.95%)
χ^2^		84.470	205.555	81.607	126.749
*p*		0.000	0.000	0.000	0.000

## Data Availability

The original contributions presented in the study are included in the article, further inquiries can be directed to the corresponding authors.

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
