# Peer review of "Epidemiology of Human Parainfluenza Virus Infections among Pediatric Patients in Hainan Island, China, 2021–2023"

_pathogens, 2024, doi:10.3390/pathogens13090740_

Round 1

Reviewer 1 Report

Comments and Suggestions for Authors

GENERAL

-          The author mentioned the “before” and “after” zero COVID policy in this paper. I suggest they add some explanation regarding this policy, including its timeline and its impact on the people.

The fact that the human parainfluenza virus is stable during the pandemic needs more explanations, especially in the discussion section. This is a very important finding

METHODS

-          I suggest the author add a few more sentences explaining Hainan Hospital

-          What were the exclusion criteria in the study?

-          What were the reasons for changing PCR into NGS?

RESULTS

-          Line 150-152: The results in this sentence were already mentioned in the first part of the Results section

-          Did the change from PCR to NGS influence the diagnosis? What were the other impacts of this change?

-          Line 171-177: The sentences were doubled

-          Line 197-201: If the zero COVID policies were working, why was the incidence of HPIV generally almost similar to before the COVID era?

-          Line 209-212: These results were already mentioned (redundancy)

DISCUSSION

-          Line 304: Was the disease increase caused by a higher incidence or a better diagnostic method?

-          Line 317-319: Did the coinfections cause prolonged stay (LOS)?

TABLES

-          Table 1: the age categories were overlapped (0-1, 1-3, 3-7, and 7-18)

-          Table 1: How did the author calculate the percentages? This may be added to the legend of the table

-          Table 1: For very small p, I suggest the author write <0.001 instead of 0.000

-          Table 1: This table needs a legend

FIGURES

-          Figure 2A: What additional information could we get from this figure? Is it necessary to display this figure?

-          A similar question was addressed in Figure 3.  

Author Response

Reviewer 1:

The author mentioned the “before” and “after” zero COVID policy in this paper. I suggest they add some explanation regarding this policy, including its timeline and its impact on the people.

Response: Thank you for your valuable feedback. We have added a detailed explanation of the Zero-COVID policy, including its timeline and impact on the population in Introduction section:

"In response to the COVID-19 outbreak in early 2020, China implemented strict public health measures, culminating in the "Dynamic Zero-COVID" policy by August 2021 to combat the Delta variant. This policy, which involved extensive lockdowns, mask mandates, and widespread testing, remained until December 2022, significantly reducing the transmission of respiratory viruses by limiting interpersonal contact. In March and August 2022, Hainan experienced two major COVID-19 waves, prompting rigorous enforcement of non-pharmaceutical interventions (NPIs) during these months, which we refer to as the "strict NPI period." These NPIs were rigorously maintained until the policy’s repeal in December 2022. The implementation of these NPIs also affected the transmission of other respiratory viruses, including respiratory syncytial virus, and potentially influenced pediatric HPIV epidemics. Analyzing HPIV infection trends before and after the pandemic could offer valuable insights for future prevention and control in children. "

The fact that the human parainfluenza virus is stable during the pandemic needs more explanations, especially in the discussion section. This is a very important finding.

Response: Thank you for highlighting this important point. We agree that the stability of the Human Parainfluenza Virus (HPIV) during the COVID-19 pandemic warrants further explanation. In the revised manuscript, we have expanded the Discussion section to provide a more in-depth analysis of this phenomenon.  In Discussion section:

"Despite the significant reduction in the transmission of other respiratory viruses during the pandemic, HPIV exhibited a notable stability in its circulation. This resilience can be attributed to several factors. First, HPIV is transmitted primarily through respiratory droplets, a mode of transmission that may have allowed it to persist even under stringent public health measures like masking and social distancing. Unlike viruses such as influenza and RSV, which saw marked declines during the pandemic, HPIV's efficient transmission mechanisms likely contributed to its continued prevalence. Additionally, HPIV might possess immune evasion capabilities that allow it to avoid detection and clearance by the host immune system, further supporting its persistence during periods of reduced human interaction. This characteristic could explain why HPIV remained relatively stable while other respiratory viruses were suppressed by NPIs. Moreover, the phenomenon of 'immunity debt'—where reduced exposure to common pathogens during the pandemic led to lower population immunity—may have made individuals, particularly children, more susceptible to HPIV infections as NPIs were lifted. Therefore, understanding the unique transmission dynamics and immune evasion strategies of HPIV is crucial for developing effective public health responses to future outbreaks. "

METHODS:

I suggest the author add a few more sentences explaining Hainan Hospital.

Response: Thank you for your insightful suggestion. We agree that providing additional information about Hainan Maternal and Child Health Hospital will enhance the readers' understanding of the study's context. We have included more details in the manuscript to explain the role and capabilities of the hospital. We have added the following sentences to the manuscript:

"Hainan Maternal and Child Health Hospital, located in Hainan Province, China, is a specialized institution dedicated to maternal and child healthcare. As one of the largest pediatric care centers on Hainan Island, the hospital is equipped with advanced diagnostic facilities, including PCR and next-generation sequencing (NGS) technologies, which were utilized in this study. The hospital serves a diverse population, providing a broad representation of pediatric respiratory infections on the island, which makes it an ideal setting for this epidemiological study."

These additions help to contextualize the study by highlighting the hospital's significant role in pediatric healthcare on Hainan Island.

What were the exclusion criteria in the study?

Response: Thank you for your question. The exclusion criteria were carefully defined to ensure the validity and reliability of our study results. The exclusion criteria are as follows:

1) Age over 18 years.

2) Outpatients (only inpatients were included in this study).

3) Patients not diagnosed with acute respiratory tract infections (ARTIs).

4) Incomplete necessary clinical information.

5) Respiratory specimens with apparent external contamination due to prolonged sampling and transportation times.

These criteria were established to ensure that the study population accurately represented the target group of pediatric patients hospitalized with ARTIs.

What were the reasons for changing PCR into NGS?

Response: Thank you for your question. The transition from PCR to next-generation sequencing (NGS) in this study was primarily driven by the need to meet evolving clinical demands. While the initial PCR method was capable of detecting 13 respiratory pathogens, the upgrade to NGS allowed for the detection of a broader range of respiratory pathogens with greater sensitivity and specificity. This change was implemented to enhance the detection capabilities, providing a more comprehensive understanding of the respiratory infections present in the study population.

RESULTS:

Line 150-152: The results in this sentence were already mentioned in the first part of the Results section

Response: Thank you for pointing out this redundancy. We have carefully reviewed the manuscript and agree that the results mentioned in lines 150-152 were previously discussed in the first part of the Results section. To avoid repetition and improve the clarity of the manuscript, we have removed the redundant sentence from lines 150-152.

Did the change from PCR to NGS influence the diagnosis? What were the other impacts of this change?

Response: Thank you for your insightful question. The transition from PCR to next-generation sequencing (NGS) did have a significant impact on the diagnostic process. The key influences and impacts of this change are as follows:

Enhanced Diagnostic Accuracy: The shift to NGS improved the sensitivity and specificity of pathogen detection, allowing for a more accurate diagnosis of respiratory infections. Unlike PCR, which was limited to detecting 13 specific respiratory pathogens, NGS provided a broader and more comprehensive analysis of the viral and bacterial landscape, leading to more precise identification of the causative agents.

Broader Pathogen Detection: NGS enabled the detection of a wider range of respiratory pathogens, including those that were not previously targeted by the PCR assays. This broader scope allowed for the identification of co-infections and rare pathogens that might have been missed with PCR alone.

Impact on Clinical Management: The increased diagnostic capability of NGS influenced clinical decision-making by providing a more complete picture of the infectious agents present in the patients. This, in turn, allowed for more tailored and effective treatment strategies, potentially improving patient outcomes.

In summary, the transition to NGS had a positive impact on the diagnostic accuracy and scope of pathogen detection, ultimately enhancing the clinical management of respiratory infections in the study population.

Line 171-177: The sentences were doubled

Response: Thank you for bringing this to our attention. We have carefully reviewed the manuscript and confirm that the sentences in lines 171-177 were inadvertently duplicated. To rectify this issue, we have removed the redundant sentences to ensure clarity and conciseness in the text.

Line 197-201: If the zero COVID policies were working, why was the incidence of HPIV generally almost similar to before the COVID era?

Response: Thank you for your insightful question. We acknowledge that the persistence of HPIV incidence at levels similar to those before the COVID era, despite the implementation of zero COVID policies, is an important observation that warrants further explanation. The explanation in Discussion section:

"Despite the significant reduction in the transmission of other respiratory viruses during the pandemic, HPIV exhibited a notable stability in its circulation. This resilience can be attributed to several factors. First, HPIV is transmitted primarily through respiratory droplets, a mode of transmission that may have allowed it to persist even under stringent public health measures like masking and social distancing. Unlike other viruses such as influenza and RSV, which saw marked declines during the pandemic, HPIV's efficient transmission mechanisms likely contributed to its continued prevalence. Additionally, HPIV might possess immune evasion capabilities that allow it to avoid detection and clearance by the host immune system, further supporting its persistence during periods of reduced human interaction. This characteristic could explain why HPIV remained relatively stable while other respiratory viruses were suppressed by NPIs. Moreover, the phenomenon of 'immunity debt'—where reduced exposure to common pathogens during the pandemic led to lower population immunity—may have made individuals, particularly children, more susceptible to HPIV infections as NPIs were lifted. Therefore, understanding the unique trans-mission dynamics and immune evasion strategies of HPIV is crucial for developing effective public health responses to future outbreaks."

Line 209-212: These results were already mentioned (redundancy)

Response: Thank you for pointing out this redundancy. We have reviewed the manuscript and agree that the results mentioned in lines 209-212 were previously discussed. To improve the clarity and conciseness of the manuscript, we have removed the redundant content.

DISCUSSION:

Line 304: Was the disease increase caused by a higher incidence or a better diagnostic method?

Response: Thank you for raising this important question. We appreciate your attention to this critical aspect of our study. In response to your comment, we have added the following explanation to the manuscript to address the potential factors contributing to the observed increase in Human Parainfluenza Virus (HPIV) incidence:

"We believe that the increase in HPIV detection rates, particularly in 2023, can be attributed to a combination of factors. Firstly, the relaxation of COVID-19 preventive measures in 2023, including a reduction in mask-wearing and an overall increase in testing, likely led to higher exposure to respiratory pathogens, resulting in more detected cases. Additionally, as the virulence of SARS-CoV-2 decreased, its restrictive influence on other respiratory pathogens, including HPIV, diminished, allowing for increased circulation of HPIV. While the transition from PCR to targeted next-generation sequencing (tNGS) may have contributed slightly to the higher detection rates, this impact was minimal, as both diagnostic methods were designed to detect all four HPIV serotypes (HPIV1-4). Therefore, the observed increase in incidence is more likely due to these external factors rather than solely improved diagnostic methods."

Line 317-319: Did the coinfections cause prolonged stay (LOS)?

Response: Thank you for your insightful comment regarding the potential impact of coinfections on the length of stay (LOS).

We conducted an analysis and found that the average length of stay was 4 days for children with single infections and 5.4 days for those with mixed infections. However, due to the difficulty in obtaining concurrent hospital occupancy data, we were unable to calculate the overall average length of stay for all patients. This limitation makes it challenging to definitively determine whether mixed infections contribute to prolonged hospitalization or extended infection duration. We recognize this as a limitation of our current study and will address this issue more comprehensively in future research.

Thank you for bringing this important aspect to our attention.

TABLES

Table 1: the age categories were overlapped (0-1, 1-3, 3-7, and 7-18)

Response: Thank you for your observation regarding the age categories in Table 1. We would like to clarify that the age groups were designed to avoid any overlap. Specifically, the age ranges are defined as follows:

0-1 refers to 0 ≤ X ≤ 1

1-3 refers to 1 < X ≤ 3

3-7 refers to 3 < X ≤ 7

7-18 refers to 7 < X ≤ 18

Thus, each category is distinct with no overlap between the age groups. We hope this clarification resolves the concern.

Table 1: How did the author calculate the percentages? This may be added to the legend of the table

Response: Thank you for your valuable feedback. The percentages in Table 1 were calculated by dividing the number of positive cases in each group by the total number of tests conducted in that group, multiplied by 100%. We have added this explanation to the legend of Table 1 for clarity.

Table 1: For very small p, I suggest the author write <0.001 instead of 0.000

Response: Thank you for your suggestion. We agreed with your recommendation and have revised the table accordingly. All instances of "0.000" in Table 1 will be changed to "<0.001" to accurately reflect the very small p-values.

Table 1: This table needs a legend

Response: Thank you for your observation. We have added a legend to Table 1 in the manuscript to provide clarity and ensure that the table is fully understood.

FIGURES:

Figure 2A: What additional information could we get from this figure? Is it necessary to display this figure? A similar question was addressed in Figure 3.  

Response: Thank you for your insightful question. We would like to clarify the distinct purposes of Figures 2A and 3 in our study:

Figure 2A: This figure is intended to highlight the patterns or differences in susceptibility to Human Parainfluenza Virus (HPIV) across different age groups of children. It provides valuable epidemiological data, allowing us to observe any age-related trends or distinctions in HPIV susceptibility, which is crucial for understanding the virus's impact on various pediatric populations.

Figure 3: This figure focuses on exploring the potential for co-infections between HPIV and other respiratory pathogens. It aims to determine whether there is a synergistic or antagonistic relationship between HPIV and other pathogens, which is important for understanding the complexity of respiratory infections and for developing appropriate treatment strategies.

Both figures address different epidemiological questions that are essential to the study, and we believe that displaying both is necessary to provide a comprehensive analysis of HPIV in the pediatric population.

Reviewer 2 Report

Comments and Suggestions for Authors

Review:

The authors provide a thorough review of the evolving epidemiology of human parainfluenza virus (HPIV) in a specific region. While similar studies have been conducted, the inclusion of regional data contributes valuable insights to the existing literature.

Specific Comments:

Lines 46–47: The authors suggest that HPIV demonstrated adaptability and coexistence during the SARS-CoV-2 pandemic. It would be beneficial to elaborate on the factors that enabled HPIV to maintain its prevalence during this period. If it does not fit within the introduction, consider including relevant data and discussion in the appropriate section.

Lines 54–57: The manuscript does not discuss the association between HPIV and croup, a significant condition linked to HPIV-1 and HPIV-2. This connection is crucial and should be addressed, possibly in the context of clinical implications.

Lines 176–177: The authors should clarify which specific months were included in each season, both in the text and in the figure legends, to ensure accurate interpretation of the seasonal data.

Line 219: The phrase "except for 3–7 year group" warrants further exploration. The authors should consider discussing potential reasons for this observation.

Lines 245–253: The manuscript refers to the colonization of S. pneumoniae and H. influenzae. Given that colonization by these microorganisms is common in children, it may be more appropriate to refer to "detection" rather than "infection." The authors might also consider discussing the implications of colonization versus infection in this context.

Lines 262–263: Additional explanation is needed to clarify the reasoning behind the observations made in this section. Expanding on potential factors or hypotheses would strengthen the argument.

Lines 305–308: The advantages of tNGS over multiplex PCR for detecting other pathogens are mentioned. It would be helpful if the authors could provide a more detailed explanation of why tNGS is superior, including specific examples or data if available.

Lines 344–345: The authors state that co-infection rates have increased. It is important to consider whether this observed increase is due to an actual rise in co-infections or merely a result of the enhanced sensitivity of the testing methods. The authors should address this point to avoid misinterpretation.

Comments on the Quality of English Language

The writtent English is easy to follow without notable flaw(s).

Author Response

The authors provide a thorough review of the evolving epidemiology of human parainfluenza virus (HPIV) in a specific region. While similar studies have been conducted, the inclusion of regional data contributes valuable insights to the existing literature.

Specific Comments:

Lines 46–47: The authors suggest that HPIV demonstrated adaptability and coexistence during the SARS-CoV-2 pandemic. It would be beneficial to elaborate on the factors that enabled HPIV to maintain its prevalence during this period. If it does not fit within the introduction, consider including relevant data and discussion in the appropriate section.

Response: Thank you for your valuable feedback on our manuscript. We appreciate your suggestion to elaborate on the factors that enabled HPIV to maintain its prevalence during the SARS-CoV-2 pandemic. In response, we have expanded on this topic in both the Introduction and Discussion sections of the manuscript. In the Introduction, we have discussed several key factors that likely contributed to HPIV's persistence during the COVID-19 pandemic.

In Introduction section:

"The ability of HPIV to maintain its prevalence during the COVID-19 pandemic may be attributed to several factors, including its transmission through respiratory droplets, which might have continued despite the widespread implementation of NPIs. These measures may not have been fully effective in preventing the spread of HPIV, particularly in environments where young children were in close contact, such as daycares and schools. Additionally, the virus's resilience in various environmental conditions, especially in tropical climates like Hainan Island, could have supported its ongoing transmission. High humidity and warm temperatures are known to influence the survival and spread of respiratory viruses. Moreover, the potential for HPIV to evade immune responses or take advantage of reduced immunity due to decreased exposure to other respiratory viruses during the pandemic may have further facilitated its continued circulation."

In Discussion section:

"The persistence and resilience of HPIV during the COVID-19 pandemic, even in the face of stringent public health measures, can be attributed to several factors. HPIV's primary mode of transmission through respiratory droplets likely allowed it to continue spreading in environments where close contact persisted, such as in households and daycare centers, despite broader social distancing measures. Unlike other respiratory viruses like influenza and RSV, which experienced marked declines, HPIV's efficient transmission mechanisms may have enabled it to maintain its prevalence. Additionally, HPIV might possess immune evasion capabilities that allow it to avoid detection and clearance by the host immune system, further supporting its persistence during periods of reduced human interaction. Moreover, the concept of "immunity debt"—where reduced exposure to common pathogens during the pandemic led to lower population immunity—may have made individuals, particularly children, more susceptible to HPIV infections as NPIs were lifted. This ability to adapt and persist underscores the importance of understanding HPIV's unique transmission dynamics and immune evasion strategies, as these characteristics could have significant implications for managing future outbreaks."

We hope these additions address your concerns and enhance the clarity and depth of our manuscript. Thank you once again for your constructive feedback.

Lines 54–57: The manuscript does not discuss the association between HPIV and croup, a significant condition linked to HPIV-1 and HPIV-2. This connection is crucial and should be addressed, possibly in the context of clinical implications.

Response: Thank you for your thoughtful and constructive feedback on our manuscript. We appreciate your observation regarding the omission of the association between HPIV and croup, particularly in relation to HPIV-1 and HPIV-2. In response, we have added a section in the Discussion:

"Moreover, it is crucial to note the clinical implications of HPIV, particularly HPIV-1 and HPIV-2, as they are closely associated with croup, a significant respiratory condition in young children. Croup is characterized by inflammation of the upper airways, leading to a barking cough and difficulty breathing, which can be severe in some cases. The association between HPIV and croup underscores the importance of early diagnosis and targeted management of HPIV infections to prevent serious outcomes, especially in pediatric populations. The continued circulation of HPIV during the pandemic, coupled with its ability to cause croup, highlights the need for heightened clinical awareness and preparedness, particularly as NPIs are lifted and respiratory viruses re-emerge. "

We believe that this addition significantly strengthens the manuscript by addressing the critical clinical implications of HPIV and providing a more comprehensive understanding of its impact on pediatric health.

Lines 176–177: The authors should clarify which specific months were included in each season, both in the text and in the figure legends, to ensure accurate interpretation of the seasonal data.

Response: Thank you for your valuable feedback. We have addressed your concern by adding the specific months included in each season both in the text and the figure legends. The seasonal distribution is now clearly defined as follows: Winter refers to December January, and February, with March to May in spring, June to August in summer, and September to November in autumn. These details have been incorporated to ensure accurate interpretation of the seasonal data.

Line 219: The phrase "except for 3–7 year group" warrants further exploration. The authors should consider discussing potential reasons for this observation.

Response: Thank you for your insightful feedback. We appreciate your attention to the phrase "except for 3–7 year group" in Line 219. In response to your comment, we have expanded the discussion to explore potential reasons for this observation.

"However, in our study, the increase in HPIV positivity rates across all age groups in 2023, except for the 3-7 years group, suggests that children in this age group may have developed partial immunity due to prior exposures, potentially offering some protection against HPIV. Additionally, this age group might have experienced more typical social interactions earlier during the pandemic recovery, which could have stabilized infection rates compared to younger children who had less exposure during critical developmental years."

Lines 245–253: The manuscript refers to the colonization of S. pneumoniae and H. influenzae. Given that colonization by these microorganisms is common in children, it may be more appropriate to refer to "detection" rather than "infection." The authors might also consider discussing the implications of colonization versus infection in this context.

Response: Thank you for your insightful feedback regarding the terminology used in our manuscript. We agree that the terms "colonization" and "infection" should be used appropriately, especially when referring to microorganisms like Streptococcus pneumoniae and Haemophilus influenzae, which are commonly found as colonizers in the respiratory tract of children.

In response to your suggestion, we have revised the manuscript to use the term "detection" instead of "infection" when referring to these microorganisms. Additionally, we have added a brief discussion on the implications of colonization versus infection in the context of our findings, emphasizing that while these pathogens were frequently detected alongside HPIV, their presence does not necessarily indicate active infection. This distinction is crucial for accurately interpreting the clinical significance of our results. The brief discussion in the Discussion section:

"The detection of Streptococcus pneumoniae and Haemophilus influenzae in co-infections with HPIV warrants a discussion on colonization versus infection. It is important to recognize that these bacteria are common colonizers of the respiratory tract in children, and their presence may not necessarily indicate active infection. Colonization can play a role in the overall respiratory health of the child, potentially influencing susceptibility to viral infections or exacerbating symptoms. Understanding whether these detections represent colonization or active infection is crucial for accurately assessing the clinical implications and guiding appropriate treatment strategies."

We believe these revisions enhance the clarity and accuracy of the manuscript. Thank you once again for your valuable input.

Lines 262–263: Additional explanation is needed to clarify the reasoning behind the observations made in this section. Expanding on potential factors or hypotheses would strengthen the argument.

Response: Thank you for your insightful comments. We have expanded the discussion section to clarify the potential factors contributing to the higher HPIV detection rates in our study compared to those reported in Southern China. The expanded content the Discussion section:

"These detection rates are higher than those reported in Southern China, which may be influenced by several factors. First, the regional differences in healthcare infrastructure and diagnostic capabilities could play a role. Hainan Island, being less developed compared to other parts of Southern China, might have experienced delayed or reduced access to healthcare services during the pandemic, leading to a buildup of cases that were later detected as the healthcare system normalized. Moreover, the unique geographic and climatic conditions of Hainan Island might have contributed to a more sustained transmission of HPIV during and after the pandemic. The island's tropical climate may facilitate year-round circulation of respiratory viruses, differing from the more seasonal patterns observed in other regions of China. These factors, combined with potential differences in population immunity due to varying levels of exposure to common pathogens during the pandemic, could explain the observed discrepancies in HPIV detection rates."

Lines 305–308: The advantages of tNGS over multiplex PCR for detecting other pathogens are mentioned. It would be helpful if the authors could provide a more detailed explanation of why tNGS is superior, including specific examples or data if available.

Response: Thank you for your insightful question. The advantages of tNGS over multiplex PCR are as follows:

  1. Broader Target Range

TNGS, as a high-throughput sequencing technique, offers a more expansive detection framework, encompassing a much broader array of targets. It can simultaneously identify hundreds to thousands of targets, whereas multiplex PCR is typically limited to fewer than a hundred targets.

  1. Enhanced Specificity

Multiplex PCR relies on locus-specific amplification, which requires high primer specificity to yield accurate results. Non-specific amplification can lead to false positives. In contrast, tNGS bases its detections on comprehensive genetic information, resulting in a lower error rate and greater accuracy.

  1. Higher Flexibility in Primer Design

TNGS is less constrained by the stringent requirements of primer design that are essential for multiplex PCR. While multiplex PCR primers must target highly conserved genomic regions to ensure specificity, tNGS can accurately distinguish between different species even when amplification is generated by homologous primers, thanks to its reliance on detailed genetic data.

To keep the manuscript concise, we have referenced these points through relevant literature rather than including the detailed explanation directly in the text. We hope this approach is understandable. Thank you again for your valuable feedback.

Lines 344–345: The authors state that co-infection rates have increased. It is important to consider whether this observed increase is due to an actual rise in co-infections or merely a result of the enhanced sensitivity of the testing methods. The authors should address this point to avoid misinterpretation.

Response: Thank you for your thoughtful feedback. We appreciate your suggestion to clarify the observed increase in co-infection rates and whether it reflects an actual rise in co-infections or is influenced by the enhanced sensitivity of the testing methods. We have added a section in the Conclusion section:

"It is also acknowledged that the rise in co-infection rates may be partly due to the enhanced sensitivity of the testing methods, particularly with the adoption of targeted next-generation sequencing (tNGS), alongside the actual epidemiological changes. "

Reviewer 3 Report

Comments and Suggestions for Authors

This study investigates epidemiological shifts in HPIV infections among children on Hainan Islands, focusing on the periods before and after the termination of the zero-COVID policy.

As the authors themselves point out, there are many limitations to the analysis, particularly changes in detection methods that undermine the reliability of the study's conclusions.

Specific comments:

1.    Result, First paragraph;

Reviewer had assumed that this paragraph described the overall characteristics of the samples collected, but it appears that this was incorrect and that it was describing the HPIV-positive samples. Are the "cases" on line 123 HPIV-positives? If so, the explanation for lines 148-153 should be moved here.

2. Table 1 should be revised to make it easier to understand. Authors should also provide overall sample characteristics in Table 1. What is the denominator of 8.32% for Male in Gender? What is the number of Males in the total sample? Similarly, Table 1 should be revised to show the sample numbers for Age and Season.

3. In Figure 1B, the figure legend should indicate which months refer to spring, summer, autumn, and winter.

3. Line 245-250, Bacteria such as Streptococcus pneumoniae and Moraxella catarrhalis are known to be bacteria that colonize the nasopharynx of healthy children. Please explain whether results of this study should lead to the treatment of these bacteria as pathogens.

4. Discussion, First paragraph;

In the introduction section, authors stated that HPIV demonstrated appeared to coexist with SARS-CoV-2 at a rate consistent with its typical seasonal patterns, despite the decrease in interpersonal contacts due to these public health measures (line 45-48). In fact, it appears that many HPIVs have been detected in Hainan in 2021 and 2022 as well. Do the authors believe that the effect of NPIs on HPIV transmission in Hainan Island is different from that of other viruses?

5. Discussion, First paragraph;

Please indicate what year and month the NPI was implemented in Hainan Island, what month the “strict NPI period (line 270)” was, and from what month it was relaxed.

6. Discussion, First paragraph;

In the introduction section, authors stated that the unique climatic and geographical attributes of Hinan Island potentially present distinct health challenges (line73-74). Was this uniqueness related to the higher detection rate than those reported in Southern China, or is Hainan Island a developing region (line264-266)?  The authors should explain their ideas more clearly so that readers can understand them, including how they relate to the NPI period.

7. Authors stated that the change in diagnostic methods over time could have influenced the detection rates of various pathogens (line 330-332). If the authors have actually performed their tNGS and multiplex PCR on HPIV-positive samples simultaneously and observed the detection limit Ct value, etc., please include this information.

Author Response

This study investigates epidemiological shifts in HPIV infections among children on Hainan Islands, focusing on the periods before and after the termination of the zero-COVID policy.

As the authors themselves point out, there are many limitations to the analysis, particularly changes in detection methods that undermine the reliability of the study's conclusions.

Specific comments:

  1. Result, First paragraph;

Reviewer had assumed that this paragraph described the overall characteristics of the samples collected, but it appears that this was incorrect and that it was describing the HPIV-positive samples. Are the "cases" on line 123 HPIV-positives? If so, the explanation for lines 148-153 should be moved here.

Response: We appreciate the reviewer's careful consideration and the opportunity to clarify this matter. The term "cases" on line 123 refers to the overall characteristics of all pediatric inpatients diagnosed with acute respiratory tract infections (ARTIs) during the study period, not just the HPIV-positive cases. This paragraph discusses the demographic details of the 19,339 pediatric patients admitted with ARTIs between January 2021 and December 2023.

The explanation on lines 148-153 specifically addresses the subset of these patients who tested positive for HPIV, totaling 1,395 cases. To avoid any potential confusion, we have revised the manuscript to clearly differentiate between the overall ARTI cases and the HPIV-positive cases. Additionally, we have moved the following sentence from Section 3.2 to Section 3.1: "A total of 19,339 pediatric patients hospitalized for acute respiratory tract infections (ARTIs) were included in the study (4,419 in 2021, 5,188 in 2022, and 9,732 in 2023), all of whom were tested for HPIV. Of these, 1,395 tested positive for HPIV."

These adjustments will enhance the clarity of the manuscript and ensure that the information is accurately presented. Thank you for your valuable feedback.

  1. Table 1 should be revised to make it easier to understand. Authors should also provide overall sample characteristics in Table 1. What is the denominator of 8.32% for Male in Gender? What is the number of Males in the total sample? Similarly, Table 1 should be revised to show the sample numbers for Age and Season.

Response: Thank you for your thorough review and valuable feedback on our manuscript. We have carefully considered your comments and made the following revisions to Table 1:

We identified an error in the calculation of the percentage for males in the Gender category. The correct value is 7.68%, and we have made this correction. Additionally, we conducted a thorough review of all data to ensure accuracy in the revised Table 1. We have also included the denominators for each percentage to make the table easier to understand.

We believe these revisions enhance the transparency and comprehensibility of the data presented in Table 1. We greatly appreciate your insightful suggestions, which have helped us improve the quality of our manuscript. Thank you again for your valuable feedback. 

  1. In Figure 1B, the figure legend should indicate which months refer to spring, summer, autumn, and winter.

Response: Thank you for your insightful suggestion. We have addressed your comment by updating the figure legend for Figure 1B to include the specific months corresponding to each season. The legend now indicates that Spring refers to March–May, Summer to June–August, Autumn to September–November, and Winter to December–February. This addition should help clarify the seasonal data presented in the figure.

  1. Line 245-250, Bacteria such as Streptococcus pneumoniaeand Moraxella catarrhalisare known to be bacteria that colonize the nasopharynx of healthy children. Please explain whether results of this study should lead to the treatment of these bacteria as pathogens.

Response: We appreciate the reviewer’s insightful comment. It is indeed recognized that Streptococcus pneumoniae and Moraxella catarrhalis are common colonizers of the nasopharynx in healthy children. However, these bacteria can act as opportunistic pathogens, particularly in the context of viral co-infections, where they may contribute to the severity of respiratory infections. In our study, the detection of these bacteria in co-infections with HPIV does not automatically imply a need for treatment as primary pathogens, but rather underscores their potential role in exacerbating disease severity when present alongside viral infections.

The clinical decision to treat these bacteria as pathogens should be based on the overall clinical presentation, including the presence of symptoms consistent with bacterial infection, laboratory findings, and the severity of the respiratory illness. In cases where bacterial co-infection is suspected to contribute significantly to the clinical condition, appropriate antibacterial therapy may be warranted. However, it is crucial to differentiate between colonization and active infection to avoid unnecessary antibiotic use.

To address this concern more clearly in our manuscript, we will add a discussion on the distinction between bacterial colonization and infection, as well as the implications for treatment in cases of co-infection with HPIV.

"It is well known that Streptococcus pneumoniae and Moraxella catarrhalis are common nasopharyngeal colonizers in children, but they can become pathogenic, especially during viral infections like HPIV. Viral-induced disruption of mucosal barriers can facilitate bacterial invasion, leading to more severe respiratory symptoms. In this study, although these bacteria were frequently detected as co-pathogens, their presence alone does not warrant antibiotic treatment unless supported by clinical evidence of bacterial infection. Overuse of antibiotics in cases of mere colonization could contribute to resistance, emphasizing the need for careful clinical assessment before initiating treatment."

  1. Discussion, First paragraph;

In the introduction section, authors stated that HPIV demonstrated appeared to coexist with SARS-CoV-2 at a rate consistent with its typical seasonal patterns, despite the decrease in interpersonal contacts due to these public health measures (line 45-48). In fact, it appears that many HPIVs have been detected in Hainan in 2021 and 2022 as well. Do the authors believe that the effect of NPIs on HPIV transmission in Hainan Island is different from that of other viruses?

 Response: Thank you for your insightful comment regarding the effect of non-pharmaceutical interventions (NPIs) on HPIV transmission on Hainan Island.

In the Introduction section, we mentioned that HPIV demonstrated adaptability and appeared to coexist with SARS-CoV-2 at a rate consistent with its typical seasonal patterns, despite the reduction in interpersonal contacts due to public health measures. However, as observed in our study, a substantial number of HPIV cases were indeed detected on Hainan Island in 2021 and 2022, which may suggest a different response to NPIs compared to other respiratory viruses.

To address your question, we believe that the effect of NPIs on HPIV transmission in Hainan Island may have differed from that of other viruses. While NPIs were effective in reducing the transmission of many respiratory viruses, HPIV appears to have exhibited resilience, maintaining or even recovering its typical seasonal pattern once NPIs were relaxed. This resilience might be attributed to HPIV's inherent viral characteristics, regional population immunity levels, or differences in how NPIs were implemented and adhered to in Hainan compared to other regions.

This observation highlights the complex interaction between public health measures and viral transmission dynamics, and it suggests that HPIV may have unique epidemiological behaviors that warrant further investigation. We appreciate your comment, as it underscores the importance of considering these nuances in our analysis and discussion.

Thank you for bringing this to our attention, and we will ensure to explore this aspect more deeply in future research.

  1. Discussion, First paragraph;

Please indicate what year and month the NPI was implemented in Hainan Island, what month the “strict NPI period (line 270)” was, and from what month it was relaxed.

 Response: Thank you for your question regarding the timeline of the Non-Pharmaceutical Interventions (NPIs) on Hainan Island. The "strict NPI period" mentioned in the manuscript specifically refers to the months of April and August 2022. We have now provided additional details in the Introduction to clarify this timeline.

"In response to the COVID-19 outbreak in early 2020, China implemented strict public health measures, culminating in the "Dynamic Zero-COVID" policy by August 2021 to combat the Delta variant. This policy, which involved extensive lockdowns, mask mandates, and widespread testing, remained until December 2022, significantly reducing the transmission of respiratory viruses by limiting interpersonal contact. In March and August 2022, Hainan experienced two major COVID-19 waves, prompting rigorous enforcement of non-pharmaceutical interventions (NPIs) during these months, which we refer to as the "strict NPI period." These NPIs were rigorously maintained until the policy’s repeal in December 2022. The implementation of these NPIs also affected the transmission of other respiratory viruses, including respiratory syncytial virus, and potentially influenced pediatric HPIV epidemics. Analyzing HPIV infection trends before and after the pandemic could offer valuable insights for future prevention and control in children."

Thank you for your valuable feedback.

  1. Discussion, First paragraph;

In the introduction section, authors stated that the unique climatic and geographical attributes of Hinan Island potentially present distinct health challenges (line73-74). Was this uniqueness related to the higher detection rate than those reported in Southern China, or is Hainan Island a developing region (line264-266)?  The authors should explain their ideas more clearly so that readers can understand them, including how they relate to the NPI period.

Response: Thank you for your insightful comments. To address your concerns and clarify the relationship between the higher detection rate of HPIV in Hainan Island compared to Southern China, we have added the following explanation to the first paragraph of the Discussion section:

"These detection rates are higher than those reported in Southern China, which may be influenced by several factors. First, regional differences in healthcare infrastructure and diagnostic capabilities could play a role. Hainan Island, being less developed compared to other parts of Southern China, might have experienced delayed or reduced access to healthcare services during the pandemic, leading to a buildup of cases that were later detected as the healthcare system normalized. Moreover, the unique geographic and climatic conditions of Hainan Island might have contributed to more sustained transmission of HPIV during and after the pandemic. The island's tropical climate may facilitate year-round circulation of respiratory viruses, differing from the more seasonal patterns observed in other regions of China. These factors, combined with potential differences in population immunity due to varying levels of exposure to common pathogens during the pandemic, could explain the observed discrepancies in HPIV detection."

We believe this addition clarifies the impact of Hainan Island's unique attributes on HPIV detection rates and their relation to the NPI period. Thank you again for your valuable feedback.

  1. Authors stated that the change in diagnostic methods over time could have influenced the detection rates of various pathogens (line 330-332). If the authors have actually performed their tNGS and multiplex PCR on HPIV-positive samples simultaneously and observed the detection limit Ct value, etc., please include this information.

Response: Thank you for your insightful question. To address your concern, we have added the following details to the Methods section:

"The generated sequencing read data underwent a bioinformatics process that included multiple steps: quality control, database alignment, microorganism identification, and the generation of a reliable pathogen report. To identify positive signals for specific pathogens, the sequence reads matching the targeted species were counted and normalized to reads per 100,000 reads (RPhK). Samples with specific RPhK values of ≥ 10 were considered positive for the presence of the corresponding pathogen, while those with lower values were reported as 'absent.'"

We hope this addition clarifies the diagnostic approach and the criteria used to determine positivity in our study. Thank you again for your valuable feedback.

Round 2

Reviewer 3 Report

Comments and Suggestions for Authors

All but one of the reviewers' comments have been satisfactorily revised by the authors.

 In the authors' response to the comments 5, there are repeated paragraphs in the discussion that should be revised. (Lines 313-327, Lines 337-352)

Author Response

Comments: In the authors' response to the comments 5, there are repeated paragraphs in the discussion that should be revised. (Lines 313-327, Lines 337-352).

Response:We sincerely appreciate the reviewer's attention to detail and their constructive feedback. We recognize that there was an unintended repetition in the discussion section of our manuscript, specifically in lines 313-327 and lines 337-352. To address this issue, we have revised these sections to eliminate redundancy while preserving the key points necessary for our discussion.

Revised Sections:

Original (Lines 313-327):

"These factors, combined with potential differences in population immunity due to varying levels of exposure to common pathogens during the pandemic, could explain the observed discrepancies in HPIV detection rates. The differences in HPIV detection rates among pediatric patients can be attributed to varying economic and healthcare conditions, with HPIV positivity rates being significantly lower in developed regions compared to developing regions. Therefore, it is essential to continue improving healthcare conditions in Hainan to address this disparity effectively. This upward trend of HPIV detection can be attributed to the relaxation of NPIs as the pandemic waned and societies returned to normal activities during 2021 to 2023. The initial suppression of HPIV transmission during the strict NPI period demonstrated the effectiveness of these measures in controlling respiratory infections. However, the subsequent increase in HPIV cases highlights the virus's resilience and ability to re-establish transmission once NPIs were lifted.

The persistence and resilience of HPIV during the COVID-19 pandemic, even in the face of stringent public health measures, can be attributed to several factors. "

Revised Version:

"The differences in HPIV detection rates among pediatric patients are likely influenced by varying economic and healthcare conditions, with positivity rates being significantly lower in developed regions compared to developing regions. As healthcare conditions improve in regions like Hainan, the disparity in detection rates may decrease. The initial suppression of HPIV transmission during the strict NPI period highlighted the effectiveness of these measures in controlling respiratory infections. However, as NPIs were relaxed during 2021 to 2023, the subsequent increase in HPIV cases illustrates the virus's resilience and ability to re-establish transmission. "

Original (Lines 337-352):

"particularly children, more susceptible to HPIV infections as NPIs were lifted. This ability to adapt and persist underscores the importance of understanding HPIV's unique transmission dynamics and immune evasion strategies, as these characteristics could have significant implications for managing future outbreaks.

Moreover, it is crucial to note the clinical implications of HPIV, particularly HPIV-1 and HPIV-2, as they are closely associated with croup, a significant respiratory condition in young children. Croup is characterized by inflammation of the upper airways, leading to a barking cough and difficulty breathing, which can be severe in some cases. The association between HPIV and croup underscores the importance of early diagnosis and targeted management of HPIV infections to prevent serious outcomes, especially in pediatric populations. The continued circulation of HPIV during the pandemic, coupled with its ability to cause croup, highlights the need for heightened clinical awareness and preparedness, particularly as NPIs are lifted and respiratory viruses re-emerge.

Despite the significant reduction in the transmission of other respiratory viruses during the pandemic, HPIV exhibited a notable stability in its circulation. This resilience can be attributed to several factors. "

Revised Version:

"particularly in children, as NPIs were lifted. This underscores the importance of understanding HPIV’s unique transmission dynamics and immune evasion strategies.    

Additionally, HPIV-1 and HPIV-2 are closely associated with croup, a significant respiratory condition in young children. The association between HPIV and croup emphasizes the importance of early diagnosis and targeted management to prevent severe outcomes in pediatric populations. The resilience of HPIV during the pandemic, despite a significant reduction in other respiratory viruses, highlights the need for continued clinical awareness and preparedness as NPIs are relaxed and respiratory viruses re-emerge."

We believe these revisions address the reviewer’s concerns and improve the clarity and cohesiveness of our discussion section.